# Plant-Derived Recombinant Vaccines against Zoonotic Viruses

**DOI:** 10.3390/life12020156

**Published:** 2022-01-21

**Authors:** Gergana Zahmanova, Katerina Takova, Rumyana Valkova, Valentina Toneva, Ivan Minkov, Anton Andonov, Georgi L. Lukov

**Affiliations:** 1Department of Plant Physiology and Molecular Biology, University of Plovdiv, 4000 Plovdiv, Bulgaria; katerina.takova@uni-plovdiv.bg (K.T.); boncheva@uni-plovdiv.bg (R.V.); toneva@plantgene.eu (V.T.); 2Center of Plant Systems Biology and Biotechnology, 4000 Plovdiv, Bulgaria; minkov@plantgene.eu; 3Institute of Molecular Biology and Biotechnologies, 4108 Markovo, Bulgaria; 4Department of Medical Microbiology and Infectious Diseases, Max Rady College of Medicine, University of Manitoba, Winnipeg, MB R3E 0J9, Canada; Anton.Andonov@umanitoba.ca; 5Faculty of Sciences, Brigham Young University–Hawaii, Laie, HI 96762, USA; georgi.lukov@byuh.edu

**Keywords:** recombinant vaccines, zoonotic viruses, virus-like particles, plant molecular farming, zoonotic influenza, emerging coronaviruses, West Nile virus, dengue virus, Zika virus, yellow fever virus, Ebola virus, Crimean–Congo hemorrhagic fever virus, Chikungunya virus, HIV, rabies virus, Hantaviruses, Henipaviruses, Newcastle virus, Hepatitis E virus

## Abstract

Emerging and re-emerging zoonotic diseases cause serious illness with billions of cases, and millions of deaths. The most effective way to restrict the spread of zoonotic viruses among humans and animals and prevent disease is vaccination. Recombinant proteins produced in plants offer an alternative approach for the development of safe, effective, inexpensive candidate vaccines. Current strategies are focused on the production of highly immunogenic structural proteins, which mimic the organizations of the native virion but lack the viral genetic material. These include chimeric viral peptides, subunit virus proteins, and virus-like particles (VLPs). The latter, with their ability to self-assemble and thus resemble the form of virus particles, are gaining traction among plant-based candidate vaccines against many infectious diseases. In this review, we summarized the main zoonotic diseases and followed the progress in using plant expression systems for the production of recombinant proteins and VLPs used in the development of plant-based vaccines against zoonotic viruses.

## 1. Introduction

Zoonoses are diseases transmitted from vertebrate animals to humans and are considered one of the most important threats to Public Health [1]. Zoonotic viruses cause illnesses with a high death rate and numerous long-term health issues. In addition, zoonotic diseases also affect the livestock sector and can have a tremendous economic impact [2]. More than 60% of emerging human infectious diseases are zoonoses and 99% of the emerging unknown viruses are zoonotic with a potentially high risk of spreading globally [3,4,5]. The recent outbreak of severe acute respiratory syndrome-associated coronavirus 2 (SARS-CoV-2), causing more than 5 million deaths worldwide by the end of October 2021, demonstrates the significance of zoonoses. Domestic and wild animals serve as reservoirs of zoonotic viruses, which are transmitted via direct or indirect contact [3,6,7]. The indirect transmission occurs by vectors (insects and arthropods), which significantly impacts disease transmission dynamics and complicates the measures taken to control zoonoses. Viruses, which have a transmission cycle between animal reservoirs where they primarily amplify, and their vectors (mosquitoes, ticks, midges) are known as arthropod-borne viruses or Arboviruses. Before transmission to a susceptible host, arboviruses must replicate in the arthropod vectors [8].

Table 1 summarizes important zoonotic viruses with details on the animal reservoirs, distribution, and fatality rate. It is worth noting that some viruses, such as HIV, began as zoonotic viruses, but because of mutations, they are no longer considered zoonotic.

Vaccination is an essential method in eradicating zoonoses and the spread of highly pathogenic viruses and is of great importance to the “Global One Health” paradigm. The Global One Health Initiative’s mission includes efforts to prevent the cross-species transmission of infectious diseases, assess environmental and social impact, and develop adequate science-based risk management policies [9]. An important measure in the One Health Initiative is the creation of widely available vaccines to be used by both humans and animals. The plant expression systems may offer a successful alternative for vaccine production compared to conventional expression systems, especially for animal vaccination (Figure 1).

## 2. Plants as an Expression System for Vaccine Production

For more than three decades, plants have been used for the production of therapeutic recombinant proteins, especially subunit vaccines, and VLP vaccines to cope with emerging and re-emerging diseases. Plants possess great potential in the production of vaccines because they are a safe, cost-effective, and scalable expression system. The achievements of plant molecular farming (PMF) and plant-based production of vaccines and diagnostic reagents have been summarized in a number of review articles [10,11,12,13,14,15,16]. The ability of plants to produce candidate vaccines has been demonstrated in several clinical trials, and two plant-derived vaccines (influenza vaccine and Newcastle disease vaccine) have been approved for commercial use [17,18,19]. Together with the licensing of a plant-derived recombinant human β-glucocerebrosidase, marketed as ELELYSO [20], these achievements promoted plant molecular farming and led to the rapid development of plant expression systems and the overcoming of some of their disadvantages. Increasing the yield [21], modifying plant glycosylation patterns [22], reducing the cost of production and purification [23], and increasing the recombinant protein stability [24,25,26] significantly improved the plant expression systems, making them competitive with the commercially used microbial expression systems. However, despite these achievements, the PMF has not been able to become a technology of choice for the production of pharmaceutical recombinant proteins. There are multiple and complex reasons for the slow progress of PMF. Among them are the strict regulation of GMO and pharma products. In addition, the industry is reluctant to restructure its fermentation infrastructure, and the fact that the productivity of plants, as compared to best practices, is lower also does not help the implementation of PMF in the production of recombinant protein pharmaceuticals [14]. Still, we may be witnessing a breakthrough thanks to the Medicago’s success in launching influenza and SARS-CoV-2 vaccines, which have undergone phase-3 clinical trials and are currently awaiting regulatory approval. As a result of the rapid production of large quantities of recombinant proteins within less than one week due to the transient expression approach, plant-based vaccines are becoming very attractive option when there is urgent need caused by the swift transmission of emerging pandemic viruses.

The advantages and disadvantages of the most commonly used systems for vaccine and therapeutic proteins production (Yeast, E. coli, insect cells, Chinese hamster ovary (CHO) cells, and embryonated hen’s eggs (EHE)), and their comparison with plant expression systems have been extensively reviewed by several authors [14,27,28,29]. Figure 2 summarizes the main characteristics of the used bio-pharming systems and highlights the advantages and disadvantages of plant expression systems.

The early concept for producing cheap and easy-to-use edible vaccines, created by Charles Arntzen [30]**,** has been studied for decades. Edible vaccines were developed, and some were used in clinical trials [31,32,33] but were later dismissed, especially for humans, mainly because of the impossibility of precise dosage of the recombinant protein within the living plant [14]. This concept of edible vaccines is still being developed for veterinary purposes [34,35,36]. Plant-made vaccines for veterinary use are very promising, mainly because of the possibility of oral delivery of the recombinant vaccine, at low cost and with a long shelf life, especially in cases where cereals are used.

Plant molecular farming is based on three main technological platforms: transient expression (generally in *Nicotiana benthamiana*), stable transgenic plants (transgenic and transplastomic), and plant cell-suspension cultures [37,38,39,40]. Transient expression in plants provides a higher yield of recombinant proteins within a shorter timeline than the stable modified plants and has been used successfully as a rapid response platform addressing emerging viral diseases such as Influenza and COVID 19. The production of recombinant proteins using transient expression based on RNA viral vectors has gained wide popularity due to numerous advantages: simple and easy engineering, high level of production of the protein of interest, scalability, lower risk, and lower costs [41]. Mainly small (+) RNA viruses such as tobacco mosaic virus (TMV) [42], cowpea mosaic virus (CMV) [43], and potato virus X (PVX) [40] are utilized as viral vectors for commercial protein production [44,45]. Earlier, the virus-based vectors expressed the gene of interest in addition to the virus’s own natural genes. Later on, the viruses were modified by removing all unessential elements for target protein expression, resulting in higher yield and better stability of the viral complex [46]. Transient expression, also known as agroinfiltration, involves the introduction of genes into plant leaves by infiltrating them with disarmed *Agrobacterium tumefaciens* carrying binary vectors [47]. The advantages of this approach have led several companies (Medicago Inc. (Québec, QC, Canada); iBio/Caliber Therapeutics (Bryan, TX, USA); Kentucky BioProcessing Inc. (KBP), Owensboro, KY, USA; Fraunhofer USA (Plymouth, MI, USA)) to invest in this plant-based platform for rapid and large-scale production of vaccines and therapeutics [48].

Stable transformation of plants takes longer, but once optimized, the transgenic system allows long-term and large-scale recombinant protein production. Various plant species such as tobacco, potato, cereals, lettuce, tomato, carrot, alfalfa, and oilseeds have been used for stable transformation by nuclear or chloroplast genome manipulation. The main approaches used to achieve stably transformed plants are Agrobacterium-mediated transformation or biolistic process [39,49]. The chloroplasts are sometimes a preferable target for stable transformation instead of the nuclei due to their high copy number in the plant cell, leading to high recombinant protein production.

The choice of technology to achieve a plant-based vaccine depends on the route of vaccine administration, the ability to achieve high levels of recombinant protein expression, and a low value of downstream processing. Edible plants (tomato, carrot, lettuce, rice, and maize) are suitable for oral administration. Cereal crops are preferable for long storage. Tobacco plants are preferable as model plants and to achieve high levels of expression.

The use of bioreactors from plant cell cultures in PMF makes it possible to overcome some of the shortcomings of transgenic crops. In vitro growing of plant cells under controlled conditions allows for the precise monitoring of cell growth and protein production and the development of good manufacturing practices. The approach of producing valuable pharmaceutical proteins in plant cell bioreactors is becoming increasingly common due to strong restrictive measures on the use of transgenic plants in the field, especially in Europe [50].

## 3. Virus-like Particles (VLPs) for Vaccine Development

VLPs are recognized as safe and effective vaccines against viral diseases [11,51]. They resemble regular viruses’ outer structure, composition, and size, but they lack genetic material, which makes them non-infectious [52]. Because VLPs retain the native antigenicity of the viruses they mimic, they can efficiently elicit cellular and humoral immune responses [53]. The virus-like particles can be formed from structural proteins belonging to one type or multiple types of viruses. Those composed of structural proteins or immunogenic epitopes from different viruses are known as chimeric VLPs [54,55,56,57]. Such VLPs can be used as immune modulators and self-adjuvants to provoke strong immune responses against the presented immunogenic epitopes [58,59,60]. The use of virus-like particles has also been recently utilized by the PMF community [54,61,62]. VLPs-based candidate vaccines expressed in plants have been developed against several zoonotic diseases and the efficacy of these vaccines is currently being evaluated. Chimeric plant-derived VLP vaccines are also being developed against diseases such as cancer, allergies, and autoimmune diseases [63].

## 4. Influenza Viruses

Influenza viruses are members of the family *Orthomyxoviridae* and include four genera (Influenza virus A, B, C, and D) [64]. Only influenza A and B are clinically significant for human health [65]. While influenza B infects only humans, influenza A infects humans and a wide variety of mammals and birds. The influenza A virus is zoonotic and can generate pandemic viruses by switching the host. Influenza is an enveloped virus with a spherical or pleomorphic shape, and is 80–120 nm in diameter. Its genome contains eight segments of single-stranded negative-sense RNA. The lipoprotein envelope of the influenza A virus contains two glycoproteins, hemagglutinin (HA), and neuraminidase (NA) [66]. After infection, HA and NA are the primary antigens inducing antibody production [67]. To date, 18 HA and 11 NA gene variants have been identified [68]. Of these variants, only four HA (H1, H2, H3, and H5) and two NA (N1 and N2) are considered as potentially serious pandemic threats [69]. HA is a stronger antigen than NA; thus, it is the main object of interest for vaccine development against influenza.

The World Health Organization (WHO) estimated that the flu kills about 250,000 to 500,000 people annually [70]. There is a shortage of influenza vaccines globally, and the ability for a “rapid response” vaccine production against pandemic influenza strains needs to improve [13]. Plant molecular farming technology can address these needs because it provides fast development and manufacturing of vaccines and scalable production. In 2009, the US Defense Advanced Research Projects Agency (DARPA) invested USD 100 million in four companies (Fraunhofer USA, the Center for Molecular Biotechnology in Delaware, Kentucky Bioprocessing, and Medicago Inc.) to produce 100 million influenza vaccines in plants in a month [71]. In 2012, Medicago produced 10 million doses of the H1N1 vaccine in a one-month “rapid fire test” [72]. They were able to achieve that due to many years of scientific development and the use of transient expression vectors [43], providing synthesis and accumulation of large quantities of recombinant proteins in plants within a week. D’Aoust et al. [73] demonstrated the production of VLPs composed only of influenza H5 by means of transient expression in *N. benthamiana.* Animal studies with low doses HA VLPs showed protective immunity in mice. The development of the candidate monovalent VLP vaccines against H5N1 pandemic strains [74,75] and the polyvalent HA VLPs for seasonal flu, which successfully passed phase-3 clinical trials [17,62,76] demonstrate the great potential plant-based technologies have in the future of vaccine production. These studies show that plant-derived HA VLPs candidate vaccines can provide protection against respiratory illnesses caused by influenza viruses in humans. Medicago Inc. has successfully completed phase-3 clinical studies for a plant-derived VLP quadrivalent flu vaccine (NCT03321968, NCT03301051, NCT03739112) [77,78]. The reported high efficacy of this plant-made flu vaccine is an important milestone in the progress of PMF.

In addition, the efficient production of hemagglutinin-based VLP vaccines in *N. benthamiana* has been demonstrated in academic studies by Rybicki et al. [13] and Smith et al. [79]. In Smith’s study, H6-subtype VLPs were transiently expressed and evaluated for efficacy in chickens against the heterologous H6N2 virus. Their findings demonstrate the potential of the plant-produced H6N2 HA vaccine for poultry. HA from various influenza strains was expressed in plants, and its immunogenicity was assessed [80,81,82,83,84,85,86,87,88,89]. Modifications of the HA structure were made to achieve a high level of recombinant protein accumulation: the sequences were optimized, the transmembrane domain and native signal peptide were removed, and an endoplasmic retention signal was inserted at the C terminus. The yield of the transiently expressed protein was HA variant dependent. The yield of H3 was 200 mg/kg of fresh weight (FW) tobacco leaves [81], while the yield of H1 was 400–1300 mg/kg FW [82]. Shoji et al. generated trimeric HA, which mimics the authentic HA structure, by introducing a trimerization motif from a heterologous protein into the HA sequence [83]. Immunization with the generated HA induced a protective immune response upon challenge of mice with a lethal viral dose [83]. Musiychuk et al. developed a plant expression system that achieved high-level target antigen expression in plants by engineering a thermostable carrier molecule fused to HA from the influenza A/Vietnam/04 (H5N1) virus [90].

Stable expressions of immunogens from the influenza virus are less used compared with transient expressions. In a review paper, Redkiewicz et al. summarized the results achieved from different plant expression systems for the production of HA [91]. Here, we will only mention some of the latest results achieved by stable expression of HA: avian H5N1 HA was found to be stably expressed in tobacco seeds [92] with a yield of 3.0 mg of the viral antigen per g of seed; and H3N2 nucleoprotein was detected in transgenic maize with a yield of 35 µg of NP/g seed [93]. The relatively low levels of recombinant protein accumulation in stable transgenic plants indicate that transient expression is more promising for developing an influenza vaccine in plants.

The feasibility of plants for the production of chimeric VLPs presenting the extracellular domain of the M2 influenza protein (M2e) as a candidate for universal influenza vaccines has also been investigated. Hepatitis B core protein (HBcAg), Hepatitis E open reading frame 2 (ORF2) capsid protein, Human papillomavirus 16 (HPV-16) L1 protein, Cowpea Mosaic Virus (CPMV), and tobacco mosaic virus (TMV) have been used as scaffolds of the M2e influenza peptide [60,94,95,96,97,98,99]. The immunogenicity of some of the chimeric VLPs presenting M2e was assessed in mice, and they often demonstrated a high immune response with protective activity [88,91,92,93]. Blokina et al. developed a chimeric protein combining the M2e and HA2 influenza A antigens with bacterial flagellin, which has adjuvant properties. The chimeric recombinant protein was expressed using a highly efficient PVX-based expression system in *N. benthamiana*, resulting in protein accumulation of up to 300 µg/g of fresh leaf tissue. Mice that were intranasally immunized with the purified chimeric protein survived a lethal challenge with the influenza A virus strain A/Aichi/2/68 (H3N2) [100].

## 5. Emerging Coronaviruses

Within the last twenty years, the emergence of three novel coronaviruses causing severe acute respiratory syndrome was observed: SARS-CoV-1, MERS-CoV, and SARS-CoV-2. They have the crown-like protruding knobs on their surface that are typical for all coronaviruses, a large, positive-strand RNA genome of approximately 30,000 nucleotides, and are classified in the genus *Betacoronavirus* [101]. Genus *B**etacoronavirus* also harbors the so-called “common cold” human coronaviruses hCoV-OC43 and hCoV-HKU [102]. The virion is composed of four structural proteins, spike (S), envelope (E), membrane (M), and nucleocapsid (N), all of which are immunogenic, but only the spike (S) protein gives rise to neutralizing antibodies [103,104]. The swift spread of the SARS-CoV-2 virus worldwide, infecting billions of people with millions of fatalities, sent an alarming message to the scientific community that there is a need for rapid development of effective vaccines against viruses such as the SARS-CoV-2. Researchers working in plant molecular farming have been prompted to analyze how plant-based production systems can meet this need [28,105,106,107]. The urgent need for a COVID-19 vaccine and diagnostic reagents has been addressed by the rapid and large-scale production of SARS-CoV-2 structural proteins in plants.

The first report of a successful plant-produced vaccine candidate against SARS-CoV-2 was from the Canadian company Medicago Inc. [108]. The strategy included transient expression in *N. benthamiana* of the modified SARS-CoV-2 S protein with stabilizing point mutations, a plant signal peptide, and the influenza HA transmembrane domain and cytoplasmic tail replacing the equivalent sequences in SARS-CoV-2. Medicago successfully produced and purified modified VLPs built from the S protein. Following positive Phase-1 results, Medicago has launched Phase-2/3 (ClinicalTrials.gov Identifier: NCT04636697) and Phase-3 clinical trials (NCT05040789) (Table 2). Recently, Medicago Inc. and GlaxoSmithKline announced a 71% efficacy rate of their plant-based vaccine against all variants of SARS-CoV-2 [109].

The American biotech companies iBio [112] and Kentucky BioProcessing (KBP) [113] also announced that they have produced subunit vaccines against SARS-CoV-2 based on plants. KBP used transient expression in *N. benthamiana* plants for the KBP-201 vaccine production. The KBP-201 with CpG oligonucleotides as an adjuvant is currently in phase-1/2 clinical trials (Table 2; ClinicalTrials.gov Identifier: NCT04473690). iBio produced a VLPs-based vaccine (IBIO-201) containing S protein linked to the LicKM^TM^ booster molecule. IBIO-201 demonstrated that it can elicit an anti-SARS-CoV-2 immune response in preclinical studies [114].

Baiya Phytopharm, a biotech company in Thailand, used *N. benthamiana* for developing a subunit-based vaccine against SARS-CoV-2. Their candidate vaccines were tested for immunogenicity in mice and monkeys. The Baiya SARS-CoV-2 Vax 1 candidate vaccine is currently in phase-1 clinical trials (Table 2; ClinicalTrials.gov Identifier: NCT04953078) [115].

The Lomonossoff group, John Innes Center, Norwich, also announced SARS-CoV-2 candidate vaccine production based on plants expressing the S, E, and M proteins. They purified “crown-shaped” particles from plant leaves infiltrated with the S protein [116]. Pogrebnyak et al. were also successful in generating S1 protein composed of the N-terminal and the receptor-binding domains in tobacco and tomato plants and observed robust immune response upon immunization of mice [117]. Another study described the expression of the S1 protein in lettuce and transplastomic tobacco. However, immunological studies have not yet been reported [118]. Zheng et al. [119] expressed the SARS-CoV N protein, and Demurtas et al. [120] expressed the M and N proteins in *N. benthamiana* and demonstrated their immunogenicity in animal models. Mamedov et al. co-expressed the N protein with the receptor-binding domain (RBD) in *N. benthamiana* to produce the SARS-CoV-2 antigens in plants. The cocktail of these two antigens elicited high-titer antibodies against SARS-CoV-2 [121]. Maharjan et al. also expressed RBD and proved that the plant-derived recombinant RBD elicited a humoral immune response in mice [122]. The overall effort to produce plant-derived SARS-CoV-2 VLPs suitable for vaccine development in a short period of time is unprecedented [123,124,125,126].

## 6. Filoviruses

### 6.1. Ebola Virus (EBOV)

EBOV causes severe hemorrhagic fever in humans and non-human primates. It has been associated with periodic outbreaks in Central Africa, and it also has the potential to be used as a bioterrorism agent [127]. The Ebola virus belongs to the family *Filoviridae*, genus *Ebolavirus* [128]. EBOV virion has a negative-sense RNA genome encoding seven gene products: four nucleocapsid proteins (NP, VP35, VP30, L); two membrane-associated proteins (VP24, VP40); and a transmembrane glycoprotein (GP) [129,130,131]. The GP mediates the viral entry process into cells and is the key immunogenic protein in vaccine protection [132,133]. In recent years, remarkable success has been achieved in developing candidate vaccines against filoviruses in non-human primate models and clinical trials [134,135,136,137]. Plant expression systems offer cost-effective alternatives to conventional vaccine production systems against EBOV. Nieto-Gómez et al. designed a multiepitopic protein called Zerola, carrying epitopes from the EBOV glycoprotein (GP) [138]. They demonstrated the ability of nuclear-transformed tobacco lines to produce Zerola antigenic protein and created a plant-based, low-cost candidate vaccine against EBOV. Phoolcharoen et al. developed a novel approach by fusing EBOV glycoprotein to the heavy chain of 6D8 mAB and co-expressing it in *N. benthamiana* [139]. After purification, this product formed immune complexes (EIC). Mice immunized subcutaneously with plant-derived EIC produced anti-EBOV antibodies at levels comparable to those obtained with GP1 virus-like particles, demonstrating the effectiveness of the plant-expressed EIC as a vaccine candidate [139,140].

### 6.2. Marburg Virus (MV)

The Marburg virus is a part of the same family as the Ebola virus, causing a highly infectious zoonotic hemorrhagic fever. It also has the potential to be used as a biological weapon. To date, there is no published evidence on the creation of a plant-made vaccine against the Marburg virus. Future efforts exploring the generation of such a vaccine can be based on and expand the current knowledge of EBOV candidate vaccines.

## 7. Bunyavirales

### 7.1. Crimean–Congo Hemorrhagic Fever Virus (CCHFV)

CCHFV causes tick-borne zoonosis in over 30 countries in Africa, Asia, and Europe [141,142]. Humans are infected mainly by tick bites, particularly from the *Hyalomma* genus, and by exposure to the tissue and blood of infected animals, which are asymptomatic [143]. The CCHFV is an enveloped single-stranded negative-sense RNA virus with a tri-segmented genome. It belongs to order *Bunyavirales*, family *Nairoviridae*, *Orthonairovirus* genus [144]. The RNA segments, S (small), M (medium), and L (large), encode the viral nucleoprotein (N), the glycoproteins Gn and Gc, and the RNA-dependent RNA polymerase [145]. Due to the lack of approved vaccines, high pathogenicity, and the increasing spread of CCHFV, the development of a vaccine and the use of plants for its production is a viable alternative. Ghiasi et al. expressed the Gn/Gc glycoproteins in transgenic tobacco leaves and hairy roots. Oral immunization of mice with transgenic leaves and roots elicited specific IgA and IgG production [146]. The use of an edible animal vaccine may reduce the risk of zoonotic transmission of CCHFV. An attractive antigen for CCHFV vaccine production is the nucleoprotein (N) [147]. To date, no expression of the N protein in plants and assessment of its immunogenicity have been reported.

### 7.2. Rift Valley Fever Virus (RVFV)

RVFV causes severe epidemics among ruminants and humans in Africa and the Arabian Peninsula [148]. In animals, RVFV is a vector-borne infection mainly transmitted by the Aedes and Culex mosquito [149,150]. Humans are often infected by close contact with infected animals and rarely through bites from infected mosquitoes [151,152]. The RFV disease symptoms range from uncomplicated acute febrile illness to severe hemorrhagic disease and death [149]. Similarly to the other arboviruses (Zika virus, Chikungunya virus, and Dengue virus), RVFV may emerge worldwide due to the widespread distribution of arthropod vectors. RVFV belongs to the family *Peribunyaviridae,* genus *Phlebovirus* [153]. The RVFV genome contains three single negative-stranded RNA, packaged into viral nucleocapsid protein (N) and enveloped by a lipid bilayer containing two viral glycoproteins (Gn/Gc) [154,155]. The main targets for vaccine production are the Gn and N proteins. The Rybicki group described the production of chimeric RVFV virus-like particles transiently expressed in *N. benthamiana* [156]. They replaced Gn’s ectodomain (Gne) and fused it to the transmembrane domain of avian influenza H5N1 HA, which allowed the chimeric VLPs to be produced and purified. The generation of Gn-HA chimeric RVFV VLPs in plants was the first demonstration of its kind to elicit a specific antibody response in mice [156]. Kalbina et al. expressed the N and Gn proteins in *Arabidopsis thaliana* and demonstrated that the orally administrated proteins are immunogenic in mice. No VLP formation has been reported [157].

### 7.3. Hantaviruses

Hantaviruses (HVs) are emerging pathogens belonging to family *Bunyaviridae* and are known for causing hantavirus pulmonary syndrome or hemorrhagic fever with renal syndrome depending on the geographical location (North America, or Europe and Asia) [158]. HVs are transmitted to humans from their natural reservoirs, rodents, through contact with infected animals and their excrements. Annually, these viruses cause approximately 200,000 infections in humans worldwide [159,160]. The Hantavirus genome comprises three segments, named small (S), medium (M), and large (L). The L segment encodes the viral polymerase, the M segment encodes the envelope glycoproteins Gn and Gc precursor (GPC), and the S segment encodes the viral nucleocapsid protein (N) [161]. The N protein and the Gn and Gc glycoproteins are promising candidates for vaccine development against HVs [159,162]. Kehm et al. generated transgenic tobacco and potato plants expressing the Hantavirus Puumala N protein and observed specific IgG and IgA immune responses upon oral or intraperitoneal immunization of rabbits [163]. Mice that were orally immunized with recombinant Puumala virus N protein did not induce an immune response. Khattak et al. demonstrated that the recombinant viral protein was completely degraded by trypsin and/or pepsin [164], which could explain the negative outcomes of the oral immunizations.

The Hepatitis B nucleocapsid tolerates the insertion of foreign sequences at its c/e1 region and was used to display several immunogenic epitopes for vaccine development in plants [56,94,98,165]. A number of research teams have used a similar approach to create chimeric HBc VLPs presenting various HV immunogenic epitopes, which have never been used in plants [166,167,168,169,170]. Geldmacher et al. successfully inserted a gene fragment encoding 120 amino acids of the N gene from two different hantavirus strains (Dobrava and Hantaan) into the hepatitis B nucleocapsid gene, resulting in a chimeric formation of VLPs exposing the inserted foreign protein segment on the surface [167]. These chimeric VLPs elicited cross-reactive antibody responses to the two HV strains [167,171]. Plant molecular farming researchers can use similar approaches for the generation and production of vaccines against HVs.

## 8. Togaviruses

### Chikungunya Virus (CHIKV)

CHIKV is an arbovirus transmitted by mosquitoes. It causes fever, joint pain, and acute polyarthritis [172,173]. Although the CHIKV disease is non-lethal and self-limiting, more than 30% of infected individuals can develop chronic joint pain [174]. In 2004 the CHIKV had a major outbreak spreading from sub-Saharan Africa and Southeast Asia [175]. The virus is classified as a member of the *Togaviridae* family, genus *Alphavirus* [176]. The RNA genome has two open reading frames (ORF); ORF2 encodes five structural proteins (capsid (CP), three envelope glycoproteins (GPs E1, E2, and E3), and 6K viroporin) [177,178]. Since there are no vaccines or treatments against the CHIKV infection, the use of plants can offer a fast and cost-effective platform for vaccine production. The CHIK-VLPs expressed in yeast and insect cells have been explored as vaccine candidates against CIKV. The CHIK-VLPs are morphologically identical with the native virion, and they induce cellular and humoral immune responses [179,180,181,182,183]. These studies were taken a step further by expressing ORF2 in plants to produce CHIK-VLPs. The Lomonossoff group produced capsid-like particles (CLPs) in *N. benthamiana* through transient expression of ORF2 or a fragment of the ORF2 coding capsid protein [116]. The results show that C proteins can be produced in plants following autoproteolytic cleavage of the expressed polyprotein. Not all envelope glycoproteins could be expressed in plants, only E2 was detected, and no structures resembling full CHIK-VLPs were observed [116].

## 9. Flaviviruses

The viruses from the genus *Flavivirus*, family *Flaviviridae*, cause many diseases transmitted to humans through vectors such as mosquitoes and ticks. West Nile virus (WNV), Dengue virus (DENV), Zika virus (ZIKV), Yellow fever virus (YFV), Japanese encephalitis virus (JEV), and Tick-Borne encephalitis virus (TBEV) are causing the most common flaviviral infections [184,185].

### 9.1. West Nile Virus (WNV)

West Nile virus is an emerging mosquito-borne flavivirus with a wide geographical distribution that includes part of Europe, Africa, the Middle East, south and central Asia, and Australia [186]. The transmission cycle of WNV is birds–mosquitos–birds, with migratory birds being primarily responsible for the virus dispersal, including the reintroduction of WNV from endemic areas into areas with sporadic outbreaks [186,187,188]. The WNV infection is usually asymptomatic (in around 80% of the infected people), or it could develop into a West Nile fever (headache, sore throat, backache, and diarrhea) or a severe West Nile disease [189,190,191]. WNV is a small (40–50 nm) spherical icosahedral virus. The outer protein shell is enveloped by a lipid membrane derived from the host cell [192]. The RNA genome is enclosed into a polyprotein that is cleaved post-translation to yield three viral structural proteins: nucleocapsid (C), pre-membrane (prM), and envelope glycoprotein (E), and seven nonstructural proteins (NSPs) [193,194,195]. The envelope glycoprotein that mediates viral entry into host cells by binding to cellular receptors is also the major antigenic determinant targeted by the host [196,197]. The WNV E protein shares a similar structure with the flavivirus E glycoproteins. With its C-terminal DIII (wDIII) domain, which contains epitopes recognized by neutralizing antibodies, the E proteins interact with cell receptors during viral transmission [198,199].

The envelope protein and its DIII domain are in the main focus for vaccines against WNV. Vaccines have been developed for horses, but there is no approved vaccine against WNV for human use. As reviewed by Filette et al. [195], the development of a cost-effective vaccine for veterinary use could be a solution to eradicate the virus spread and to control WNF disease in humans and animals. Therefore, many research groups explored making WNV recombinant vaccines made in plants as a safe and cost-effective means of production. The Qiang Chen research group from Arizona State University has a number of candidate vaccines developed against WNV based on the DIII domain of the E protein (wEDIII) [200]. They demonstrated that plants successfully produced wEDIII domains with an average level of 100 µg/g leaf fresh weight. The plant-derived wEDIII protein was immunogenic and potentially protective in mice [201,202,203]. Chen et al. also produced the entire E protein- and prM-E-based WNV VLPs in plants. Only the results of sucrose gradient centrifugation suggested the assembly of prM-E VLPs [51]. They also fused wEDIII to HBcAg, and produced chimeric VLPs, which elicited strong B and T-cell responses against WNV in mice [204].

The latest study of the Rybicki group reported the development of chimeric AP205 phage virus-like particles displaying WNV envelope EDIII domains. They used the SpyTag/SpyCatcher (ST/SC) conjugation system [205] to generate a virus-like particle-display-based vaccine candidate in *N. benthamiana*. A quantity of 5 μg of purified chimeric VLPs subcutaneously applied in mice elicited a potent antibody response to WNV EDIII [206].

### 9.2. Dengue Virus (DENV)

Dengue Virus is an emerging arbovirus that is distributed in tropical and subtropical areas of the world by mosquitos (*Aedes aegypti* and *A. albopictus*). The WHO estimates that worldwide, more than 3.6 billion people are at risk of being infected by DENV. Of them, annually, more than 200 million will be infected, with 21,000 fatalities [207]. Dengue is an acute febrile viral disease that may proceed asymptomatically or result in self-limited dengue fever, but it could also escalate into severe dengue hemorrhagic fever and dengue shock syndrome [208]. Therefore, the development of a scalable and safe vaccine is needed, and plants may offer the solution. Similar to all other Flaviviruses, DENV’s genome is made up of a positive-sense single strand RNA, which has a single ORF encoding all structural and non-structural proteins into one polyprotein. The polyprotein is then proteolytically cleaved into three structural proteins (capsid, precursor membrane, and envelope) and at least seven non-structural (NS) proteins [209]. DENV has four antigenically distinct serotypes (DENV-1-4) [210]. The dengue DIII domain of the E protein (dEDIII) can induce neutralizing antibody responses and is the main target for vaccine development [211,212]. There are several examples of expression of dEDIII from different serotypes and prM in plants [213,214,215,216,217,218]. dEDIII was expressed in *N. tabacum* by stable nuclear [219] or chloroplast transformation [216]. Transient expression of dEDIII in *N. benthamiana* leads to higher recombinant protein quantities than stable expression, 0.28% [212,219] vs. 0.13 to 0.25% [216], respectively [212,216,219]. The development of oral vaccination through the expression of tetravalent EDIII in lettuce chloroplasts was explored by van Eerde et al., showing that EDIII-1-4 can induce an immunogenic effect in rabbits [217].

Martínez et al. [215] used a strategy to display epitopes of dEDIII on the surface of HBcAg capsid-like particles. Using the same strategy, Pang et al. [220] inserted a consensus sequence of dEDIII into the immunodominant c/e1 loop region of tandem HBcAg and demonstrated the production of chimeric VLPs (tHBcAg-cEDIII) in *N. benthamiana* [165]. A recent study conducted by the Lomonossoff group (Ponndorf et al. [221]) described the production of DENV VLPs in *N. benthamiana*. Using the transient co-expression of DENV structural proteins (SP) and truncated versions of the non-structural proteins (NSPs), they successfully purified DENV VLPs. Immunogenicity assays revealed that the plant-made DENV VLPs led to a high antibody response in mice.

Existing Ebola recombinant immune complex (RIC) technology [139] was used by Kim et al. to develop a Dengue RIC technology (DERIC). A dEDIII gene fragment was fused with the heavy chain of 6D8 mAB to prepare a chimeric Dengue–Ebola recombinant immune complex. DERIC induced a strong virus-neutralizing anti-cEDIII humoral immune response without using adjuvants in subcutaneously immunized mice and showed the potential to be a candidate vaccine [218].

### 9.3. Zika Virus (ZIKV)

The vector for Zika virus transmission is primarily *Aedes aegypti* mosquito, albeit several other species are also involved (*A. africanus*, *A. albopictus*, *A. hensilli*) [222]. Zika virus is also transmitted through sexual contact, blood transfusions, and from mother to fetus during pregnancy [223]. The greatest danger for humans is the vertical transmission during pregnancy, resulting in the development of fetal microcephaly [224]. In 2016, the WHO declared the Zika virus a “public health emergency of international concern” when Brazil reported an association between Zika virus infection and microcephaly [225]. ZIKV infections in humans are usually asymptomatic or characterized by fever and cutaneous rash [226]. Two genotypes of ZIKV are distinguished, African and Asian [227]. ZIKV is a Flavivirus with a genome organization and virion structure similar to the other flaviviruses. The flaviviruses’ genetic similarity can provoke the antibody-dependent enhancement of infection (ADE), and thus, can present challenges for vaccine development [228,229]. The major virion surface glycoprotein E is responsible for attachment to cellular receptors, membrane binding and fusion for viral entry, and mediating viral assembly [230]. The zE glycoprotein, with its DIII domain, has great potential to induce potent neutralizing antibodies, making it a principal candidate for a subunit vaccine [231]. Chen et al. developed a zE-based candidate subunit vaccine by means of transient expression in *N. benthamiana* plants. The plant-derived envelope protein elicited potent zE-specific antibodies and cellular immune responses that correlate with protective immunity against the Zika virus in mice [232]. Chen et al. also devised VLPs carriers based on the HBcAg that presented zDIII, showing that chimeric VLPs can be easily purified in large quantities from plants and can elicit a potent humoral and cellular immune response in mice [233]. Diamos et al. demonstrated that recombinant immune complexes (RIC) are a potent platform to improve the immunogenicity of weak antigens by fusing a ZIKV envelope domain III (ZEIII) to the IgG heavy chain (N-RIC) [234]. In mice, the IgG fusions elicited up to 150-fold high titers of Zika-specific antibodies, compared to the ZEIII antigen alone, which neutralized ZIKV using only two doses without an adjuvant [234].

Interestingly, codelivery of hepatitis B nucleocapsid/zEDIII VLPs with RIC zEDIII was capable of producing a synergistic effect in terms of enhancing the immune response [235].

The Lomonossoff group transiently expressed ZIKV prM-E proteins in *N. benthamiana* and showed that plants successfully expressed and accumulated E proteins. The protein yield was low, which led to limited VLP purification due to poor VLP formation [116].

### 9.4. Yellow Fever Virus (YFV)

The mosquito-transmitted yellow fever virus causes high fatality outbreaks in South America and Africa [236]. According to the WHO, annually, YFV causes 200,000 severe cases of yellow fever (YF) and up to 60,000 deaths [236,237]. The RNA genome of YFV encodes seven non-structural (NS1, NS2A, NS2B, NS3, NS4A, NS4B, and NS5) and three structural proteins (envelope (E), membrane (M), and capsid (C)) [238]. Although there is an effective live attenuated vaccine against YFV (YF-Vax^®^, Sanofi-Pasteur, Lyon, France, and 17DD-YFV, Bio-Manguinhos, Rio de Janeiro, Brazil), the increased demand and reports of severe reactions associated with the live vaccines have provoked interest in the development of safer subunit vaccines against YFV.

Two different constructs, (a) the gene coding for the envelope (E) protein alone or (b) a fusion of E with enzyme lichenase gene (YFE LicKM), were transiently expressed in *N. benthamiana*. Both plant-derived proteins, YFE and YFE-LicKM, elicited virus-neutralizing antibodies in mice and protected over 70% of them from a lethal challenge infection [239]. Currently, there are no reports describing the development of a safe and efficacious vaccine against YFV in plants using the VLP approach. In their review, Hansen et al. describe, in detail, the different platforms for creating a vaccine against YFV [240].

### 9.5. Japanese Encephalitis Virus (JEV)

Japanese encephalitis is an arbovirus that causes 45,000 human encephalitis cases and 10,000 deaths globally [241]. The primary amplifying hosts are pigs, while water birds are carriers and mosquitoes are vectors [185]. JEV is an enveloped virus with a positive single-stranded RNA genome. The JEV genome has a single ORF encoding three structural proteins (C, prM, and E), and seven non-structural proteins (NS1, NS2A, NS2B, NS3, NS4A, NS4B, and NS5) [242]. The envelope protein (E) is the primary target of neutralizing antibodies. The recombinant JEV E protein, expressed in the baculovirus expression system, induced a neutralizing antibody response and protective immunity against a lethal challenge with JEV [243]. Plants’ expression systems have been used successfully for the production of the E surface glycoprotein. Wang et al. developed transgenic rice expressing the E protein with a yield of 1.1–1.9 μg/mg of total soluble protein. Intraperitoneal administration in mice of plant-derived E protein elicited specific neutralizing antibodies. In addition, also in mice, the oral administration of the E protein induced a mucosal immune response [244]. Chen et al. used a different approach. They created chimeric virus-like particles (CVPs) based on the bamboo mosaic virus (BaMV) to present domain III of the E protein. The CVPs were found to elicit neutralizing antibodies against JEV infection in mice [245].

### 9.6. Tick-Borne Encephalitis Virus (TBEV)

Tick-Borne encephalitis virus is transmitted by *Ixodes* spp ticks and causes a potentially fatal neurological infection known as tick-borne encephalitis (TBE). The virus is endemic to Europe, Siberia, northern China, Japan, and South Korea [246]. Each year, more than 10,000 cases of TBE are registered, of which 3000 in Europe, and in terms of morbidity, this frequency is second only to JEV among the neurotropic flaviviruses [247,248]. TBEV has the same genome organization as the other Flaviviruses. The TBEV genome in one open reading frame codes for the expression of three structural (C, prM, and E) and seven non-structural proteins. Specifically, for TBEV, prM and E form smaller, capsid-less virus-like icosahedral particles. These particles do not have the structural protein C and RNA, and they are non-infectious [247]. The recombinant prM and E VLPs and virion-derived particulate immunogens are highly immunogenic and can protect animals. In addition, the M/E protein has also been used in diagnostic assays [249,250,251].

No studies have been reported on the M and E expression in plants, with the exception of the announced project “Development of edible vaccines against Tick-Borne encephalitis” [252] of Örebro University, Sweden.

## 10. Human Immunodeficiency Viruses (HIV)

Human immunodeficiency virus (HIV) is classified as a member of the *Lentivirus* genus of the *Retroviridae* family [253,254]. Phylogenetical analysis showed close simian relatives of HIV-1 in chimpanzees [255] and sooty mangabeys [256], demonstrating the zoonotic origin of this virus [257]. Genetically, there are two types of HIV (HIV-1 and HIV-2). Of the two, HEV-1 has spread rapidly worldwide, and it is the primary cause of the global HIV pandemic [258].

Vaccine development against HIV is still a challenge due to the hyper-variability of the viral envelope (Env) glycoprotein, which results in immune evasion [259]. To address the global needs for HIV prevention, plants represent a viable option as a protein expression system [13,260,261].

A promising approach for an HIV vaccine is the development of artificial proteins based on conservative T-cell and B-cell epitopes. Notably, the gp41 and gp120 regions of HIV-1 are crucial to virus function and immune protection, making them valuable targets for vaccine investigation [262]. Multiple polyvalent HIV proteins have elicited immunological responses in animals (Table 3). For example, the C4(V3)6 protein, based on gp120, expressed in lettuce, induced an immune response in mice [263]. The p24 protein, expressed transiently or via transplastomic or transgenic transformation, induced a humoral immune response [264,265,266]. A small epitope from gp41 displayed on the surface of CPMV stimulated an antibody response in mice [267]. HBsAg has also been utilized as a carrier for HIV polyproteins, providing excellent immunogenic characteristics while being stably expressed in *N. benthamiana* [268,269]. Soluble gp140 was produced and successfully co-expressed with a human chaperone protein to achieve higher yields of the viral protein [270]. gp40, produced via transient expression in *N. benthamiana*, formed a trimeric structure and elicited a robust immunological response in rabbits [271]. Despite the substantial number of studies on this topic, none of the HIV plant-based vaccine candidates have reached the clinical trial stages.

## 11. Rabdoviruses

### Rabies Virus (RABV)

Rabies virus is a neurotropic agent that causes acute infection of the central nervous system (rabies) in mammals. This can be transmitted to humans by bites of infected animals. The host range predominantly includes carnivores and bats. In humans, mortality is nearly 100% once clinical symptoms of rabies appear [285,286]. Mortality is much lower for the animals that carry the rabies virus; 14% of dogs survive, and bats can survive too [287]. The viral RNA genome encodes five viral proteins: the nucleoprotein (N), phosphoprotein (P), matrix protein (M), glycoprotein (G), and a large RNA polymerase protein (L) [288]. The surface-exposed G protein is responsible for the induction of virus-neutralizing antibodies. There are numerous examples of candidate vaccines based on the G protein or nucleoprotein produced in plants. Many review articles discuss the progress in the development of plant-based vaccines against RABV [13,116,289,290]. Transgenic tobacco plants have been modeled to produce rabies G protein at 0.38% of the total soluble leaf protein, which, upon purification and immunization, provided complete protection in mice after a lethal challenge [291]. Since tobacco cannot be used as an edible vaccine, efforts have been made to use edible plants as potential alternatives for a Rhabdovirus vaccine. Tomato [292,293,294], carrot [295], spinach [296,297], and corn [298] have been used to produce edible vaccines against rabies. Serological studies have shown that oral administration of plant-produced viral proteins caused antigen-stimulated IgG and IgA synthesis and improved the overall health of mice intranasally infected with an attenuated rabies virus [296]. Parenteral immunization of mice with purified N protein produced in transgenic tomato achieved partial protection, while oral immunization with the same product failed [293]. Orally administrated G protein (50 µg in transgenic corn) induced viral neutralizing antibodies and protected 100% of the treated mice against a challenge [298,299]. Loza-Rubio et al. demonstrated that the degree of protection achieved after 2 mg of G protein administrated orally was comparable to that conferred by a commercial vaccine [299]. This study demonstrates that cereals are an attractive platform for oral vaccine production due to the relatively high recombinant protein accumulation and the possibility for long-term storage under ambient conditions.

Oral vaccination with the plant-produced chimeric peptides, containing antigenic determinants from glycoprotein G and nucleoprotein N fused with the alfalfa mosaic virus coat protein (AlMV CP), improved weight gain in mice following a challenge with an attenuated virus strain [296]. Yusibov et al. described the expression of a similar chimeric protein (epitopes from G and N fused with the AlMV coat protein) in tobacco and spinach plants. Studies in human volunteers showed a specific response against the rabies antigens after ingesting raw spinach leaves infected with the recombinant virus [297].

## 12. Hepatitis E Virus

Hepatitis E virus (HEV) HEV is a positive-sense, single-stranded quasi-enveloped RNA icosahedral virus from the *Hepeviridae* [300]. Genus *Orthohepevirus* unite all virus isolates from mammals and birds organized into eight genotypes. Genotypes 1 and 2 are specific for humans, while 3–8 are spread mainly in animals. The main animal reservoir of HEV is the domestic swine, but other animals such as wild boar, dear, camel, etc. can be reservoirs as well [301,302,303,304,305].

HEV ORF2 encodes the viral capsid protein, which is a major immunogenic factor, and for that reason, it is of great interest for anti-HEV vaccine development [306]. Earlier PMF studies were based either on nuclear or plastid stable transformations [307,308,309]. The E2 fragment (394–607 aa) from ORF2 was cloned into a plasmid and successfully transformed into tomato plants using *Agrobacterium tumefaciens*. Although normal immunoactivity of the recombinant protein was detected, the stable nuclear transformation resulted in low accumulation of the desired product—61.22 ng/g fresh weight (FW) in fruits and 6.37–47.9 ng/g FW in leaves—which makes the methodology unsuitable for scalable production [307]. In another study, the E2 fragment was cloned into a plastid-targeting vector and delivered in tobacco via biolistic particle bombardment. Application of the plastid transformation approach resulted in an increased yield of 13.27 μg/g FW.

The generated recombinant protein was able to elicit a positive antibody response in mice [309]. In the pursuit of an oral vaccine, transgenic potatoes were developed by Agrobacterium-mediated delivery of two truncated ORF2 fragments, truncated 111N and 111N/54C. A yield up to 30 μg/g FW was observed with very few VLPs. The absence of an immune response in mice highlighted the importance of VLP formation for the induction of a proper immune response [308].

The transient expression approach was used in conjunction with the highly productive vectors pEAQ-HT and pEff, based on the Cowpea mosaic virus (CPMV) and the potato virus X (PVX), respectively. When a truncated ORF2 (110–610 aa) sequence, cloned into the pEAQ-HT vector, was expressed, it resulted in a protein production rate of 100 µg/g of FW [310], whereas the same sequence cloned into the pEff vector yielded up to 200 µg/g of FW [311,312]. Both products confirmed that the truncated HEV capsid proteins expressed in plants are suitable for diagnostic purposes [310,312]. VLP formation was also examined by atomic force and electron microscopy. The collected data confirmed self-assembled VLP particles with very heterogeneous sizes [95,312]. The immunogenicity of the plant-produced HEV 110–610 protein was examined in mice, and it was concluded that immunizations with the recombinant protein induced high titers of specific IgG antibodies in comparison to the negative control group [311]. Additionally, HBcAg was used to present an HEV immunogenic epitope. The amino acids 551–607 of the HEV ORF2 capsid protein were successfully inserted into HBV nucleocapsid gene and upon expression in plants, chimeric “ragged” VLPs were produced, which reacted with HEV antibodies [56].

## 13. Newcastle Disease Virus (NDV)

The NDV belongs to the *Paramyxoviridae* family and its single-stranded, negative sense RNA genome encodes six structural proteins: nucleocapsid (N), phosphoprotein (P), matrix protein (M), fusion protein (F), haemagglutinin-neuraminidase protein (HN), and large polymerase protein (L) [313]. The surface glycoproteins F and HN are the major immunogenic proteins [314,315] and targets for vaccine design. NDV is highly infectious and often fatal in birds, including domestic poultry [316]. Humans can be infected by NDV through direct contact with infected poultry, which is usually presented as conjunctivitis. No human cases of Newcastle disease have occurred as a result of consuming poultry products [317]. ND severely impacts the poultry industry worldwide because it causes significant morbidity and mortality [318]. To control spreading, prophylactic and emergency vaccination against ND is applied on a large scale in many countries. Various plant systems have been used for NDV vaccine production, focused primarily on the expression of the F and HN glycoproteins (Table 4).

Table 4 shows that the main research efforts are aimed at creating an oral vaccine against NDV that can be easily administered through food and can be stored long-term under ambient conditions. Immunological studies with seed-based orally administrated vaccines induced a robust immune response. Guerrero-Andrade et al. showed that 100% of the chicken fed with transgenic maize expressing the F glycoprotein survived after a lethal challenge with NDV [321]. These studies show the feasibility of edible vaccines for veterinary application.

A few years back, Dow AgroSciences made a breakthrough in plant-based vaccines. In 2016, they received the first FDA approval for the plant-derived injection application vaccine against NDV, based on HN protein produced in an *N. benthamiana* cell culture [328].

## 14. Henipaviruses

Hendra virus (HeV) and Nipah virus (NiV) have recently emerged as zoonotic pathogens affecting domestic animals (horses, pigs) with a documented spillover in humans, sometimes causing lethal disease [318,320]. Based on their genomic organization, they are currently classified in the family *Paramyxoviridae* [329]. Fruit bats of the *Pteropus* genus were confirmed as the main zoonotic hosts of Henipaviruses [330]. HeV was initially identified in an outbreak in horses in Australia. NiV was subsequently discovered in an outbreak in pigs in Malaysia [329]. Human cases have been observed after close contact with infected animals (horses and pigs). Food-borne transmission was also reported in people who consumed fruit or palm sap from containers contaminated by fruit bats [331].

HeV and NiV are single-stranded RNA, enveloped viruses. The Henipavirus genome encodes six structural proteins: nucleoprotein (N), phosphoprotein (P), matrix protein (M), fusion glycoprotein (F), attachment glycoprotein (G), and the polymerase protein (L) [332]. The F and G glycoproteins are the main immunogenic proteins and target for vaccine development. Progress in the development of vaccines against the Nipah and Hendra viruses was reported in a review article [333].

Currently, there is no published evidence suggesting the development of a vaccine against Henipaviruses in plants, which presents PMF scientists with an opportunity to make rapid progress in the development of vaccines against HeV and NiV in plants, following the achievements of the plant-based vaccine against the Newcastle disease.

## 15. Concluding Remarks

The 21st Century has seen remarkable progress in the development of plant-based vaccines against viral diseases. Although the vast majority of these are centered on aca-demic development and are still at the level of pre-clinical trials in animal models, we are also witnessing the significant achievements of vendors such as Medicago Inc and Kentucky Bioprocessing Inc, whose vaccine candidates against influenza and SARS-CoV-2 are likely to be licensed next year. Several factors are contributing to the observed progress in plant-based vaccine development. From a technical perspective, there is a shift from stable expression of viral proteins in transgenic plants to transient expression in *N. benthamiana*. The latter is much faster and can produce a decent yield of viral antigens within a week after infiltrating plant leaf material with an Agrobacterium suspension containing the target gene, offering the opportunity for a rapid response to the annual emergence of antigenically novel influenza strains or unexpected pandemics. Another very important factor for the current advancement of plant-based vaccines is the recent achievements in plant platforms that not only express target antigens but also facilitate the assembly of VLPs. The advantage of the plant-derived VLPs over simple expression of recombinant proteins in plants is that the former captures the antigenic epitopes in their native conformation, which results in enhanced immunogenicity and ultimately superior efficiency as a potential vaccine. Apart from the above-mentioned technological factors, the potential success of a Medicago plant-derived SARS-CoV-2 vaccine will accelerate the development of other plant vaccines in general.

Fischer and Buyel 2020 [14] describe four factors that affect the selection of plant expression systems: time-to-market factors; the amount of time needed for research and development (R&D); scalability; and regulatory approval. Of the four, the time-to-market factors are the biggest determinant in the selection process. In addition, while plants offer an advantage during R&D (faster transient expression), the production upscaling of plant-produced proteins is usually a challenge. However, the approval of the new plant-based vaccines is likely to open new avenues for improved manufacturing.

The majority of the zoonotic viruses listed in this review disproportionally affect developing countries. The development of new vaccines comes with a cost, which is a challenge for resource-limited countries that need these vaccines most. Plant-based vaccines carry the promise not only to be efficient in preventing zoonotic disease but also, importantly, to be cost-efficient and affordable.

Sustainable long-term cooperation and partnerships with global organizations (the WHO, the World Bank, and others) and governments must be established if plant molecular farming is to become a successful mainstream platform for vaccine generation and production.

## Figures and Tables

**Figure 1 life-12-00156-f001:**
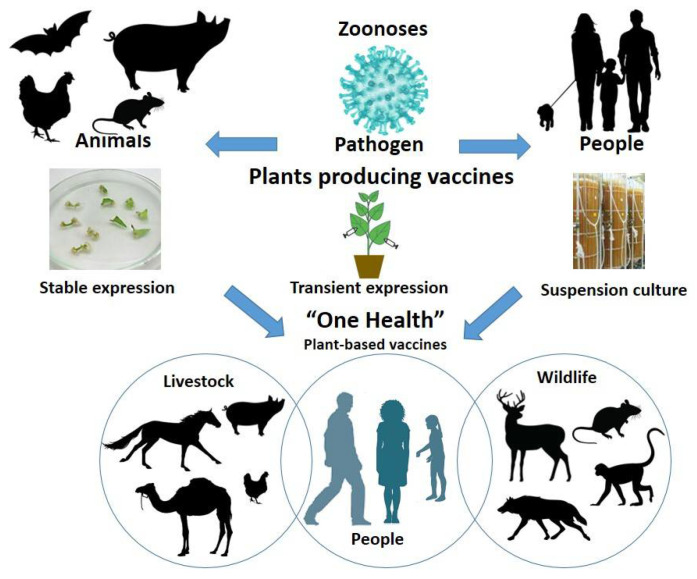
Schematic representation of the transmission of zoonotic diseases and the used plant-based production technologies (stable, transient, and suspension cultures) for recombinant vaccine production.

**Figure 2 life-12-00156-f002:**
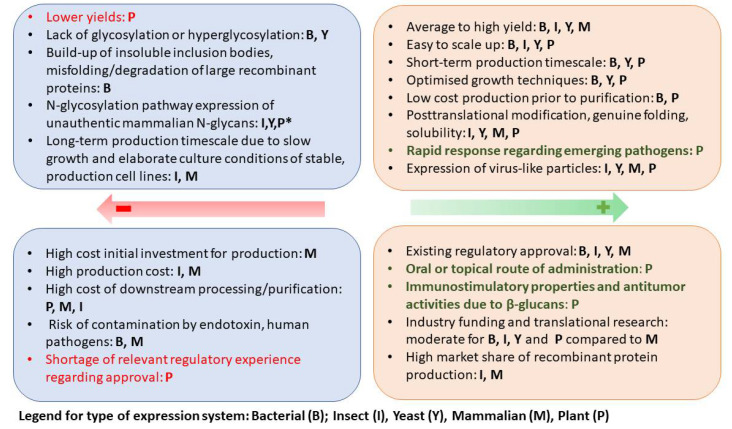
Comparison between plant expression systems and conventional expression systems. * N-linked glycans in humans differ from plant glycans, and the latter’s strong immunostimulatory effect may cause plant-derived therapeutics to have adverse events; however, these same properties are beneficial for vaccines as they enhance immunogenicity.

**Table 1 life-12-00156-t001:** Partial list of important viral zoonoses.

Virus Family/Name	Main Animal Reservoir	Distribution	Cases/Mortality Rate
Family *Orthomyxoviridae* Influenza A virus	Wide variety of birds and mammals (dogs, cats, pigs, whales, horses)	Worldwide	3–5 million severe cases annually, ~2% fatal rate
Family *Coronaviridae* Coronavirus SARS-CoV MERS CoV SARS-CoV-2	Bats, pangolins, camels, monkeys, ferrets, minks, pigs, horses, dogs, cats, snakes	Worldwide	SARS-CoV~9.6%; MERS-CoV~34.3%; SARS-CoV-2 ranging from 2% to 7.6% depending on the country
Family *Flaviviridae* West Nile virus	Birds and horses	Africa, Europe, the Middle East, North America, and West Asia	~1% of neuroinvasive disease and 5% of them are fatal
Zika virus	Human and non-human primates	Spread from Africa to the Pacific and the Americas	In Brazil, the estimated case fatality rate is 8.3%
Japanese encephalitis virus	Ardeid wading birds and pigs	Isolated in Japan. Spread to Southeast Asia, Australia	35,000–50,000 cases of JE, with a mortality rate of 10,000–15,000 people per year
Dengue virus	Monkeys and dogs	Caribbean, Central Africa, the Eastern Mediterranean, Southeast Asia, and the Western Pacific	390 million dengue virus infections per year, 1.3% fatality rate
Yellow fever virus	Humans and non-human primates	West, central and east Africa and in South America	200,000 cases of YFVD and 30,000 deaths annually
Tick-borne encephalitis virus	Small rodents and large woodland animals	Europe, Siberia, northern China, Japan, and South Korea	10,000–12,000 clinical cases of TBE and mortality rate depending on subtype (1% to 20%)
Family *Filoviridae* Ebola virus	Bats, Humans and non-human primates, wild antelopes	West and Central Africa	2013–2016—28,616 cases. The fatality is ranging from 27% to 79%
Marburg virus	Fruit bats and monkeys	Central and South Africa	Fatality ratio of up to 88%
Family *Nairoviridae* Crime-Congo Hemorrhagic Fever Virus	Wild and domestic animals such as cattle, sheep, and goats	Africa, Asia, and Europe	Fatality ratio (10–40%)
Family *Peribunyaviridae* Rift Valley fever virus	Buffaloes, camels, cattle, goats, and sheep	Sub-Saharan Africa and Arabian Peninsula	Most human cases are mild. 50% mortality within hemorrhagic form
Family *Togaviridae* Chikungunya virus	Monkeys, birds, and rodents	Southeast Asia, Europe, Caribbean, North, Central, and South America	1 million cases per year, the hospital mortality rate raging (10–26%)
Family *Rhabdoviridae* Rabies virus	Dogs and cats, foxes, wolves, mongooses, skunks, raccoons, bats	95% of cases occurring in Africa and Asia	59,000 human deaths annually
Family *Retroviridae* HIV	Originated in non-human primates	Worldwide	37.7 million people are with HIV, the fatality rate is 1.8%
Family *Paramyxoviridae* Henipaviruses (Nipah virus and Hendra henipavirus)	Large fruit bats, pigs, horses	Southeast Asia, Australia, Papua New Guinea	The fatality rate is estimated at 40% to 75%
Newcastle disease virus	Poultry and wild birds		Mild systemic ND can be observed in humans
Family *Hepeviridae* Hepatitis E virus	Pigs, wild boar, rats, rabbits, dears, dromedary camel, birds	Worldwide	3.3 million symptomatic cases with a fatality rate up to 4% in the general population
Family *Hantaviridae* Hantavirus	Rodents, bats, and insectivores	Asia, Europe, and the Americas	Mortality rates of 12%

**Table 2 life-12-00156-t002:** Clinical trial studies with the plant-based vaccine against the COVID-19 disease [110,111].

NCT Number	Study Title	Phase	Responsible Party
NCT04450004	Safety, Tolerability and Immunogenicity of a Coronavirus-Like Particle COVID-19 Vaccine in Adults Aged 18–55 Years	1	Medicago
NCT04636697	Study of a Recombinant Coronavirus-Like Particle COVID-19 Vaccine in Adults	2/3	Medicago
NCT05040789	Phase-3 Study to Evaluate the Lot Consistency of a Recombinant Coronavirus-Like Particle COVID-19 Vaccine	3	Medicago
NCT05065619	Safety Immunogenicity Study of MT-2766 in Japanese Adults (COVID-19)	1/2	Medicago
NCT04473690	KBP-201 COVID-19 Vaccine Trial in Healthy Volunteers	1/2	KBP
NCT04953078	A Study to Evaluate Safety, Tolerability, and Reactogenicity of an RBD-Fc-based Vaccine to Prevent COVID-19	1	Baiya Phytopharm

**Table 3 life-12-00156-t003:** Examples of plant-produced HIV proteins that have shown immunogenic effects in animals.

Antigen	Production System and Expression Level	Immunogenicity	References
22 aa epitope from gp41	Cowpea; CPMV-HIV chimera; n/a.	Mice; Stimulated a strong serum neutralizing antibody response in three strains of mice.	[267,272,273]
CTB-P1	*N. benthamiana* (transient); >80% of the CTB and CTB-P1 were assembled into functional oligomers.	Mice; Serum IgG response; mucosal IGA response; induction of immunological memory.	[274]
p24	*N. benthamiana* (transient); 100 μg/g FW.	Rabbit; Induced specific humoral immune response.	[266]
NV and GAG	Tomato (transgenic); VLPs composed of the major antigenic protein for the hepatitis B virus (HBV); 0.3 ng/mg powder.	Mice; Induced a humoral immune response.	[275]
polHIV-1.op	*Nicotiana tabacum* (transgenic)*;* VLPs composed of the major antigenic protein for the hepatitis B virus (HBV); 2–26 ng/g FW.	Mice; Elicited anti-HIV-1 specific CD8+ T cell activation detectable in mesenteric lymph nodes.	[268,269]
Tat	Tomato (transgenic); 1 μg/mg fruit (dry weight).	Mice; A strong anti-Tat immunological response after either intraperitoneal, intramuscular, or oral application.	[276]
p17/p24	*Nicotiana tabacum* (transient); >1 mg p24/kg of FW.	Mice; Induced humoral and T cell immune response.	[277]
CTB-MPR	*N. benthamiana* (transgenic); 1–2 mg/kg FW.	Mice; Induction of serum and mucosal antibodies.	[278]
p24	*Nicotiana tabacum* L. (transplastomic); n/a.	Mice; Serum IgG response.	[265]
C4(V3)6	Lettuce (transgenic); 240 μg/g freeze-dried leaves.	Mice; Induction of humoral and cell-mediated immune response.	[263]
p24	*Arabidopsis thaliana* (transgenic); 0.2 μg–0.5 μg/g FW.	Mice; Serum IgG response.	[264,279]
C4V3 (Multi-HIV)	Tobacco (transplastomic); 16 μg/g FW.	Mice; Induction of systemic mucosal, humoral, and T cell immune response.	[263,280,281]
Poly HIV	*Physcomitrella patens* (transgenic); 3.7 µg/g^−1^ FW.	Mice; Induction of specific antibody response.	[282]
Dgp41 and Gag	*N. benthamiana* (transient and transgenic respectively); ~9 mg/kg FW and ~22 mg/kg FW, respectively.	Mice; Serum antibodies against both the Gag and gp41 antigens were produced. CD4 and CD8 T cell response.	[283,284]
gp140	*N. benthamiana* (transient); 21.5 mg/kg FW.	Rabbits; High titers of binding antibodies, including against the V1V2 loop region, and neutralizing antibodies against Tier 1 viruses.	[270,271]

**Table 4 life-12-00156-t004:** Plant-based candidate vaccines against Newcastle disease virus.

Antigen	Production System and Expression Level	Immunogenicity	References
Fusion (F) and haemagglutinin-neuraminidase (HN) protein	Potato/stable transformation/0.3–0.6 µg/mg of total leaf protein	Oral and intraperitoneal delivery of the antigens elicited mucosal and systemic immune response	[319]
Fusion (F)	Rice/stable transformation/0.25–0.55 μg of purified NDV in 100 μg of total soluble leaf or seed proteins	Intraperitoneally immunized mice with crude protein extracts from transgenic rice plants elicited specific antibodies	[320]
Fusion (F)	Maize/stable transformation/ 0.9–3% TSP	Orally immunized chicken developed protective immune response	[321]
Fusion (F) and hemagglutinin-neuraminidase (HN) proteins	0.5–0.8% of total seed protein Mize/stable transformation	Induced specific immune response after oral administration to chicken	[322]
Fusion (F) and hemagglutinin-neuraminidase (HN) proteins	Transgenic Canola Seeds	Chickens immunized orally with recombinant HN-F showed a significant rise in specific and hemagglutination inhibition (HI) antibodies	[323]
F and HN epitopes fused to Cucumber mosaic virus (CMV)	*N. benthamiana*/CMV formed VLPs, which served as carriers for display of neutralizing epitopes of F and HN/20–200 g of infected leaves	No data	[324,325]
Hemagglutinin-neuraminidase (HN)	*N. benthamiana/*transient expression	No data	[326]
Hemagglutinin-neuraminidase protein	Transgenic *N. tobaccum* 0.069% of TSP	Orally immunized chickens developed low titers of anti-HN serum IgG	[327]
Ectodomain of hemagglutinin-neuraminidase protein	Tobacco cell culture 0.2–0.4% of TSP	Mice receiving purified eHN protein from transgenic tobacco BY-2 cells produced specific anti-NDV antibodies	[19]

## Data Availability

Not applicable.

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
