# Peer review of "Plant-Derived Recombinant Vaccines against Zoonotic Viruses"

_life, 2022, doi:10.3390/life12020156_

Round 1

Reviewer 1 Report

In this review, Zahmanova et al, addressed the “plant-derived recombinant vaccines against zoonotic viruses”. Having examined the manuscript, I note that though it discusses some interesting observations, to be considered for MDPI Life (ISSN 2075-1729), the following are some of the comments that the authors might find useful for future submission. This review is addressing some important zoonotic infections and the development of plant-based vaccines against these infections. This type of review is extremely valuable at the community and global level.

General comments

  1. The title is appropriately captured and the manuscript is well written.
  2. It would be more interesting to the readers if the future prospective of plant-derived recombinant vaccines against zoonotic viruses could be incorporated and their contribution to ultimate disease elimination.
  3. The authors should also point out the current limitations of plant-derived recombinant vaccines against zoonotic viruses.
  4. Lots of typographical errors are observed throughout the manuscript.

For example:

(i). In table 1, the authors should correct the spelling of tick-borne infections.

(ii). Page number 10, line number 384 needs to be corrected

(iii). Page number 14, line number 577 needs to be corrected (HIV-1)

(iv). Page number 16, line number 638 needs to be corrected

(v). Page number 17, table 5, authors should correct the spellings of rice and maize

  1. In page number 1 (keywords), line number 27. (Dengue virus) is repeated.
  2. In page number 18, line number 709 needs to be corrected.
  3. The binomial scientific names should always be italicized. The authors should correct the following line numbers

           (i). 339

           (ii). 482

           (iii). 513

           (iv). 652

Author Response

15th January 2022

Dear Reviewer 1,

Thank you for taking the time to review our manuscript “ Plant-derived recombinant vaccines against zoonotic viruses ” and for providing such thorough and useful feedback. We are grateful to have had the opportunity to address the comments and to submit an updated manuscript for your consideration.

We provide detail below of how we have addressed each of your comments in turn. , including updated line numbers for the relevant parts of the manuscript. We hope that you find our revisions satisfactory and look forward to receiving your decision.

Kind regards,

Dr Gergana Zahmanova

General comments:

  1. Comment #1: The title is appropriately captured and the manuscript is well written.

Response: Thank you!

  1. Comment #2: It would be more interesting to the readers if the future prospective of plant-derived recombinant vaccines against zoonotic viruses could be incorporated and their contribution to ultimate disease elimination.

Response: The wide distribution of zoonotic viruses and their variability due to the adaptive RNA genome lead to the impossibility of their complete elimination. The benefits of plant-based vaccines are that they can reduce the disease in animal vectors, and hence significantly reduce its transmission to humans. Production of recombinant proteins in plants with low cost and options for oral delivery is of great importance to the ‘Global One Health’ paradigm. We believe that this opinion is expressed in the article. However, we have also reiterated the usefulness of plant vaccines as the most fitting protection tool in the face of urgent response to new emerging threats; lines93; Still, we may be witnessing a breakthrough thanks to the Medicago’s success in launching influenza and SARS-COV-2 vaccines which have undergone clinical trials phase III and are currently awaiting regulatory approval. As a result of the rapid pro-duction of large quantities of recombinant proteins within less than a week time due to the transient expression approach, plant-based vaccines are becoming very attractive option when there is urgent need caused by swift transmission of pandemic emerging viruses.

  1. Comment #3: The authors should also point out the current limitations of plant-derived recombinant vaccines against zoonotic viruses.

Response: Text has been added to the paragraph 2. Plants as an expression system for vaccine production

 lines 86: However, despite these achievements, the PMF has not been able to become a technol-ogy of choice for the production of pharmaceutical recombinant proteins. There are multiple and complex reasons for the slow progress of PMF. Among them are the strict regulation of GMO and pharma products. In addition, the industry is reluctant to re-structure its fermentation infrastructure, and the fact that the productivity of plants compared to best practices is lower also does not help the implementation of PMF in the production of recombinant protein pharmaceuticals.  [14].

Comment #4; Lots of typographical errors are observed throughout the manuscript.

Response: All typos were corrected.

For example:

(i). In table 1, the authors should correct the spelling of tick-borne infections.

Response: line ….tick-borne

(ii). Page number 10, line number 384 needs to be corrected

Lines463; Since there are no vaccines or treatments against the CHIKV infection, the use of plats can be a fast and cost-effective platform for vaccine production.

(iii). Page number 14, line number 577 needs to be corrected (HIV-1)

Lines 664 HIV-1

(iv). Page number 16, line number 638 needs to be corrected

Lines735;.

(v). Page number 17, table 5, authors should correct the spellings of rice and maize

 Corected!

In page number 1 (keywords), line number 27. (Dengue virus) is repeated.

It was deleted.

In page number 18, line number 709 needs to be corrected.

The binomial scientific names should always be italicized. The authors should correct the following line numbers

They were corrected!

           (i). 339 lines 441

           (ii). 482

lines 565

           (iii). 513

lines 600 N. benthamiana

           (iv). 652

lines 749; Agrobacterium tumefaciens

Reviewer 2 Report

Manuscript ID: life-1508103

This manuscript describes a well performed and extended review on Plant-derived vaccines against especially zoonotic viral infections. The paper summarizes many examples of plant molecular farming and plant-based production of vaccines against a variety of emerging and re-emerging viral infections. The review is acceptable for publication, but what I miss is a direct comparison of the plant-based vaccine platform with other vaccine platforms where the antigen of interest is expressed via vectors like for instance Adeno, mRNA ‘s etc. I think that this might be quite challenging due to the large variety of parameters but, is it possible to make a table where most common vaccine platforms (in general), can be compared with plant-derived platform regarding their yield in production, stability, time scale/frame and cost of production and purification?

Author Response

Dear Reviewer 2,

Thank you for taking the time to review our manuscript “ Plant-derived recombinant vaccines against zoonotic viruses ” and for providing such thorough and useful feedback. We are grateful to have had the opportunity to address the comments and to submit an updated manuscript for your consideration.

We provide detail below of how we have addressed each of your comments in turn. , including updated line numbers for the relevant parts of the manuscript. We hope that you find our revisions satisfactory and look forward to receiving your decision.

Kind regards,

Dr Gergana Zahmanova

  1. Comment #1: The review is acceptable for publication, but what I miss is a direct comparison of the plant-based vaccine platform with other vaccine platforms where the antigen of interest is expressed via vectors like for instance Adeno, mRNA ‘s etc. I think that this might be quite challenging due to the large variety of parameters but, is it possible to make a table where most common vaccine platforms (in general), can be compared with plant-derived platform regarding their yield in production, stability, time scale/frame and cost of production and purification?

Response:

We have introduced a diagram comparing the different traditional expression systems all of which are used as a production platform for therapeutics or vaccines. Direct comparison with the newer Adeno or mRNA vaccines may not be fair as we will be comparing very different things; the latter are not only a production platform, but perhaps even more importantly a very different delivery system affecting the immune response.

P5, L109

Figure 2. Comparison between plant expression systems and conventional expression systems. *N-linked glycans in humans differ from plant glycans and the latter having strong immunostimulatory effect may cause adverse events by plant-derived therapeutics, however these same properties are beneficial for vaccines as they enhance immunogenicity.

Reviewer 3 Report

In this manuscript, the authors did a literature review for Plant-derived recombinant vaccines against zoonotic viruses. Manuscripts have much scope for improvement. For the betterment of this manuscript, I have a few suggestions for the authors, which I have penned down below.

  1. Combine concluding remarks and future direction. It would be interesting to include an estimation of production costs and yields, comparing plants with other expression systems and how realistic it is to think of an oral plant-based vaccine considering how feasible the commercialization would be.
  2. A major drawback of this literature review is the lack of any graphical representation. Therefore, please draw some nice graphical represented images that can justify this review.
  3. Clinical trials for plant-based Covid vaccine are done by many players, not only by Medicago. Please explain all in table 3. Then, refer to the WHO site for more detailed insight.
  4. In the emerging coronavirus section, the authors would advise including new covid variants as a table and infection and death rates.
  5. Write italic for Arabidopsis thaliana all over the manuscript, especially at L339 and N. benthamiana at L498. L515.
  6. The manuscript has missed many recent plants-based vaccine studies in this paper, such as: A. A Comprehensive Overview on the Production of Vaccines in Plant-Based Expression Systems and the Scope of Plant Biotechnology to Combat against SARS-CoV-2 Virus Pandemics. B. Recent Advances in the Use of Plant Virus-Like Particles as Vaccines and many more. Please include those so that your review should be more comprehensive.
  1. I found many plagiarized stuffs in this manuscript at the later part of this manuscript at L193-194, L223-226, L229-230, L264-267, L278-279, L281-283, L292-295, L343-345, L347-348, L354, L355-356, L357-358, L367-369, L370-371, L380-382, L409-411, L451-452, L453, L476-477, L481-483, L486-487, L510-511, L528-529, L531-532, L571-572, L580-582, L616-617, L623-624, L633-634, L679-680, L682-685, L709-710. Authors have picked up entire table 2 from “Plant-based vaccine research development against viral diseases with emphasis on Ebola virus disease: A review study.” However, therein table two, many studies, why authors selectively picked only 3. Also, clean it. It should not be as it is. Paper Producing Vaccines against Enveloped Viruses in Plants: Making the Impossible, Difficult have 14% similarity, and Plant-based vaccines against viruses have 6% similarity with this study. Please take care of that.

Author Response

Dear Reviewer 3,

Thank you for taking the time to review our manuscript “ Plant-derived recombinant vaccines against zoonotic viruses ” and for providing such thorough and useful feedback. We are grateful to have had the opportunity to address the comments and to submit an updated manuscript for your consideration.

We provide detail below of how we have addressed each of your comments in turn. , including updated line numbers for the relevant parts of the manuscript. We hope that you find our revisions satisfactory and look forward to receiving your decision.

Kind regards,

Dr. Gergana Zahmanova

Comment #1: Combine concluding remarks and future direction. It would be interesting to include an estimation of production costs and yields, comparing plants with other expression systems and how realistic it is to think of an oral plant-based vaccine considering how feasible the commercialization would be.

Response: We combined concluding remarks and future direction, and include the text: Lines 92: Still, we may be witnessing a breakthrough thanks to the Medicago’s success in launching influenza and SARS-COV-2 vaccines which have undergone clinical trials phase III and are currently awaiting regulatory approval. As a result of the rapid production of large quantities of recombinant proteins within less than a week time due to the transient expression approach, plant-based vaccines are becoming very attractive option when there is urgent need caused by swift transmission of pandemic emerging viruses.

Comment #2. A major drawback of this literature review is the lack of any graphical representation. Therefore, please draw some nice graphical represented images that can justify this review.

Response: p4 Figure 1 was included: Figure 1. Schematic representation of the transmission of zoonotic diseases and the used plant-based production technologies (stable, transient, and suspension cultures) for recombinant vaccines production.

and  Figure 2. Comparison between plant expression systems and conventional expression systems.

Comment #3. Clinical trials for plant-based Covid vaccine are done by many players, not only by Medicago. Please explain all in table 3. Then, refer to the WHO site for more detailed insight.

Response: p Table 3 was modified and all clinical trial studies with the plant-based vaccine against the Covid-19 disease were included.

NCT Number

Study Title

   Phase

Responsible part

NCT04450004

Safety, Tolerability and Immunogenicity of a Coronavirus-Like Particle COVID-19 Vaccine in Adults Aged 18-55 Years

    1

Medicago

NCT04636697

Study of a Recombinant Coronavirus-Like Particle COVID-19 Vaccine in Adults

    2/3

Medicago

NCT05040789

Phase 3 Study to Evaluate the Lot Consistency of a Recombinant Coronavirus-Like Particle COVID-19 Vaccine

3    3

Medicago

NCT05065619

Safety Immunogenicity Study of MT-2766 in Japanese Adults(COVID-19)

    1/2

Medicago

NCT04473690

KBP-201 COVID-19 Vaccine Trial in Healthy Volunteers

1/2

KBP

NCT04953078

A Study to Evaluate Safety, Tolerability, and Reactogenicity of an RBD-Fc-based Vaccine to Prevent COVID-19

1

Baiya Phytopharm

Comment #4. In the emerging coronavirus section, the authors would advise including new covid variants as a table and infection and death rates.

Answer 4: Respectfully, we believe that describing the different coronavirus variants (alpha, beta, gamma, delta or Omicron), and their details of infection features and death rates are outside of the scope of this review and the currently presented data in that subsection, albeit basic are adequate. 

Comment # 5. Write italic for Arabidopsis thaliana all over the manuscript, especially at L339 and N. benthamiana at L498. L515.

Answer 5: Thank you for pointing out this omission.

lines375 Arabidopsis thaliana

L498. L515. N. benthamiana

Comment # 6 The manuscript has missed many recent plants-based vaccine studies in this paper, such as: A. A Comprehensive Overview on the Production of Vaccines in Plant-Based Expression Systems and the Scope of Plant Biotechnology to Combat against SARS-CoV-2 Virus Pandemics. B. Recent Advances in the Use of Plant Virus-Like Particles as Vaccines and many more. Please include those so that your review should be more comprehensive.

Answer 6: The references were included.

lines 172. The chimeric plant-derived VLPs vaccines are also being developed against diseases such as cancer, allergies and autoimmune diseases [63].  

lines 310 - Тhe overall efforts to produce plant-derived SARS-CoV-2 VLPs suitable for vaccine development in a short period of time is unprecedented [123–126].

Comment # 7 I found many plagiarized stuffs in this manuscript at the later part of this manuscript at L193-194, L223-226, L229-230, L264-267, L278-279, L281-283, L292-295, L343-345, L347-348, L354, L355-356, L357-358, L367-369, L370-371, L380-382, L409-411, L451-452, L453, L476-477, L481-483, L486-487, L510-511, L528-529, L531-532, L571-572, L580-582, L616-617, L623-624, L633-634, L679-680, L682-685, L709-710. Authors have picked up entire table 2 from “Plant-based vaccine research development against viral diseases with emphasis on Ebola virus disease: A review study.” However, therein table two, many studies, why authors selectively picked only 3. Also, clean it. It should not be as it is. Paper Producing Vaccines against Enveloped Viruses in Plants: Making the Impossible, Difficult have 14% similarity, and Plant-based vaccines against viruses have 6% similarity with this study. Please take care of that.

Answer 7: Thank you for highlighting the plagiarized sentences. We rewrite or removed them.

L178 – Table 2 was deleted.   The information was include in L209-211 - Medicago Inc. has successfully completed phase 3 clinical studies for a plant-derived VLP quadrivalent flu vaccine (NCT03321968, NCT03301051, NCT03739112) [74,75].

L193-194 – ‘’Immunization of mice with trimeric HA elicited serum hemagglutination inhibition antibodies and protected mice against a lethal viral challenge.’  was modified: L226-Immunization with the generated HA induced a protective immune response upon challenge of mice with a lethal viral dose

L223-226 was changed to L255- Within the last twenty years the emergence of three novel coronaviruses causing severe acute respiratory syndrome was observed; SARS-CoV1, MERS-CoV and SARS-CoV2. They have the typical for all coronaviruses crown-like protruding knobs on their surface, a large, positive-strand RNA genome of approximately 30,000 nucleotides and are classified in the genus Betacoronavirus.

L229-230 – Modified to   L260- The virion is composed of four structural proteins: spike (S), envelope (E), membrane (M), and nucleocapsid (N), all of which are immunogenic, but only the spike (S) protein give rise to neutralizing antibodies

L264-267 Modified to L 297- They purified  particles with a “crown-shaped”  structure.

L278-279 and L281-283  modified to:L317- The Ebola virus belongs to the family Filoviridae, genus Ebolavirus [128]. EBOV virion has a negative-sense RNA genome encoding seven gene products: four nucleocapsid proteins (NP, VP35, VP30, L); two membrane associated proteins (VP24, VP40); and a transmembrane glycoprotein (GP).

L292-295 was modified to L328: Phoolcharoen et al. developed a novel approach by fusing EBOV glycoprotein to the heavy chain of 6D8 mAB and co-expressing it in N. benthamiana [139]. After purifica-tion this product formed immune complexes (EIC). Mice immunized subcutaneously with plant-derived EIC produced anti-EBOV antibodies at levels comparable to those obtained with GP1 virus-like particles, demonstrating the effectiveness of the plant-expressed EIC as a vaccine candidate [139,140].

L343-345 modified to  Hantaviruses (HVs) are emerging pathogens belonging to family Bunyaviridae and are known for causing hantavirus pulmonary syndrome or hemorrhagic fever with renal syndrome depending on the geographical location (North America, or Europe and Asia)

L347-348 was modified to L383;The Hantavirus genome is comprised of three segments, named small (S), medium (M), and large (L). The L segment encodes the viral polymerase, the M segment encodes the envelope glycoproteins Gn and Gc precursor (GPC), and the S segment encodes the viral nucleocapsid protein (N) [161]

L354,L355-356, It was deleted.

L357-358, modified to L388;Kehm et al. generated recently described the development of transgenic tobacco and potato plants expressing the Hantavirus Puumala N protein and observed specific IgG and IgA immune response upon oral or intraperitoneal immunization of rabbits . When administered intraperitoneally or orally to rabbits, the hantaviral recombinant N proteins obtained from transgenic tobacco and potato plants elicited specific hu-moral and mucosal immune responses

L367-369, L370-371,Modified to L398 Geldmacher et al. successfully inserted a gene fragment encoding 120 amino acids of the N gene from two different hantavirus strains (Dobrava and Hantaan) into hepati-tis B nucleocapsid gene resulting in a chimeric formation of VLPs exposing the inserted foreign protein segment on the surface [167]. These chimeric VLPs elicited cross-reactive antibody response to the two HV strains [167,171].

 L380-382, Modified to L412 The virus is classified as a member of Togaviridae family, genus Alphavirus [176]. The RNA genome has two open reading frames (ORF), the ORF2 encodes five structural proteins (capsid (CP), three envelope glycoproteins (GPs -E1, E2, E3), and 6K viropor-in) [177,178].

L409-411, deleted

L451-452, L453, Modified to Similar to all other Flaviviruses, DENV’s genome is made up of a positive-sense single strand RNA which has a single ORF encoding all structural and non-srtuctural proteins into one polyprotein.

L476-477,modified to L502 dEDIII gene fragment was fused with the heavy chain of 6D8 mAB to prepare a chimeric Dengue-Ebola recombinant immune complex. DERIC induced a strong virus-neutralizing anti-cEDIII humoral immune response without using adjuvants in subcutaneously immunized mice and has the potential to be a candidate vaccine [218].

 L481-483, L486-487, Modified to L508; The vector for Zika virus transmission is primarily Aedes aegypti mosquito, al-beight several other species are also involved (Ae. africanus, Ae. albopictus, Ae. hensilli¬) [222]. Zika virus is also transmitted through sexual contact, blood transfusions, and from mother to fetus during pregnancy

L510-511, ModifiedL535; Interestingly, codelivery of hepatitis B nucleocapsid/zEDIII VLPs with RIC zEDIII were capable of producing a synergistic effect in terms of enhancing the immune re-sponse In a later study, when co-administered to mice, the Hepatitis B capsid protein HBcAg-based zEDIII VLPs were found to behave synergistically with the recombinant immune complexes (RIC) consisting of plant-produced zEDIII fused to anti-ZIKV an-tibodies [235].

L531-532 and L528-529, Modified to L552: Two different constructs, (a) the gene coding for the envelope (E) protein alone or (b) a fusion of E with enzyme lichenase gene (YFE LicKM) were transiently expressed in N. bentamiana. Both plant-derived proteins, YFE and YFE-LicKM, elicited virus-neutralizing antibodies in mice and protected over 70% of them from a lethal challenge infection

,

L571-572, Modified to L594; Human immunodeficiency virus (HIV) is classified as a member of the Lentivirus genus of the Retroviridae family [253,254]. Phylogenetical analysis showed close simian rela-tives of HIV-1 in chimpanzees [255] and sooty mangabeys [256], demonstrating the zoonotic origin of this virus [257]

L580-582, Modified to L600: Vaccine development against HIV is still a challenge due to the hyper-variability of the viral envelope (Env) glycoprotein which results in immune evasion [259]. To address the global needs for HIV prevention, plants represent a viable option as a pro-tein expression system [13,260,261].

 L616-617, Transgenic tobacco plants have been modeled to produce rabies G protein at 0.38% of the total soluble leaf protein which upon purification and immunization provided complete protection in mice after a lethal challenge [291]

L623-624, L639 ; Serological studies have shown that oral administration of plant-produced viral pro-teins caused antigen-stimulated IgG and IgA synthesis and improved the overall health of mice intranasally infected with an attenuated rabies virus [296]. Parenteral immunization of mice with purified N protein produced in transgenuc tomato achieve partial protection, while oral immunization with the same product failed [293]

L633-634, Modified to L651: Oral vaccination with the plant-produced chimeric peptides, containing antigenic determinants from glycoprotein G and nucleoprotein N fused with the alfalfa mosaic virus coat protein (AlMV CP) improved weight gain in mice following a challenge with an attenuated virus strain [296]. Yusibov et al. described the expression of a simi-lar chimeric protein (epitopes from G and N fused with the AlMV coat protein) in to-bacco and spinach plants. Studies in human volunteers showed a specific response against the rabies antigens after ingesting raw spinach leaves infected with the recom-binant virus [297]

L679-680, Modified to L694:The NDV belongs to Paramyxoviridae family and its single stranded, negative sense RNA genome encodes six structural proteins: nucleocapsid (N), phosphoprotein (P), matrix protein (M), fusion protein (F), haemagglutinin-neuraminidase protein (HN), and large polymerase protein (L) [313].

 L682-685,  Modified to L698: The NDV belongs to Paramyxoviridae family and its single stranded, negative sense RNA genome encodes six structural proteins: nucleocapsid (N), phosphoprotein (P), matrix protein (M), fusion protein (F), haemagglutinin-neuraminidase protein (HN), and large polymerase protein (L) [313].

L709-710. Modified to L725: Hendra virus (HeV) and Nipah virus (NiV have recently emerged as zoonotic pathogens affecting domestic animals (horses, pigs) with a documented spillover in humans, sometimes causing lethal disease [318, 320]. Based on their genomic organi-zation they are currently classified in family Paramyxoviridae

Round 2

Reviewer 3 Report

I am happy with the authors response and the manuscript can be accepted in its current format. Congratulation.